# THE IMITATION GAME: TURING MACHINE IMITATOR IS LENGTH GENERALIZABLE REASONER

**Zhouqi Hua**[1,2*]   **Wenwei Zhang**[2*†]   **Chengqi Lyu**[2]   **Yuzhe Gu**[2,3]
**Songyang Gao**[2]   **Kuikun Liu**[2]   **Dahua Lin**[2,4]   **Kai Chen**[2†]

[1]Fudan University   [2]Shanghai AI Laboratory   [3]Shanghai Jiao Tong University
[4]MMLab, The Chinese University of Hong Kong
{huazhouqi,zhangwenwei,chenkai}@pjlab.org.cn

## ABSTRACT

Length generalization, the ability to solve problems of longer sequences than those observed during training, poses a core challenge of Transformer-based large language models (LLMs). Although prior work explores data-driven methods for specific arithmetic or symbolic tasks, such approaches remain task-specific with limited performance. To pursue universality, this paper studies *computable* reasoning problems, *i.e.*, those solvable by algorithms and hence by Turing machines with unbounded input length. From this perspective, this paper proposes **T**uring m**A**chine **I**mitation **L**earning (**TAIL**), which uses computer programs to directly synthesize chain-of-thought (CoT) data that imitate the execution process of a Turing machine, which *linearly* expands the reasoning steps into *atomic* states to alleviate shortcut pattern learning and explicit *memory* fetch mechanism to reduce the difficulties of dynamic and long-range data access. To validate the universality and reliability of TAIL, we build a challenging synthetic dataset spanning 8 algorithm classes and 18 tasks. Using marginally shorter CoT than reasoning models such as DeepSeek-R1 (**671B**), TAIL achieves improved length generalization and superior task performance on Qwen2.5 (**7B**) with only synthetic training data. Experiments show that Turing-machine principles, rather than human-like thinking styles, are critical for length generalization, inducing readwrite behaviors in attention layers consistent with Turing machines.

## 1 INTRODUCTION

Length generalization (Press et al., 2021), *i.e.*, the ability to handle a problem with input sequences of various lengths in the open world, especially those *longer* than previously seen, is a fundamental aspect of human intelligence and serves as a crucial evaluation criterion for AI systems (Anil et al., 2022; Sinha et al., 2024; Ahuja & Mansouri, 2024; Shi et al., 2022). Although the ability and generalizability of large language models (LLMs) to solve complex problems have been significantly improved by chain-of-thought (CoT) (Wei et al., 2022), recent studies (Saparov & He, 2022; Anil et al., 2022; Zhou et al., 2024) indicate that LLMs still struggle with length generalization, which sometimes explores and falls into shortcuts that eventually cause errors (Saparov et al., 2024).

To address the challenge, existing works (Zhou et al., 2024; 2023; Lee et al., 2023; Shen et al., 2023; McLeish et al., 2024) primarily focus on data-driven approaches, which refine the training data by modifying the structure of CoT to be more effective and generalizable. However, these methods remain inherently task-specific, *e.g.*, Index Hint (Zhou et al., 2024; 2023) for symbolic reasoning tasks and Reversed Format (Lee et al., 2023; Shen et al., 2023; Zhou et al., 2023; McLeish et al., 2024) for arithmetic problems, and yield only moderate performance gains. Thus, a question arises: *Is there a universal and effective CoT structure for length generalization?*

This paper aims to answer this question by first taking a deeper look at the commonalities among the problems. Notably, we observe that many of these tasks admit well-defined stepwise procedures that can be solved by program algorithms that generalize to inputs of arbitrary length. We refer to

---

* Equal contribution, † Corresponding author

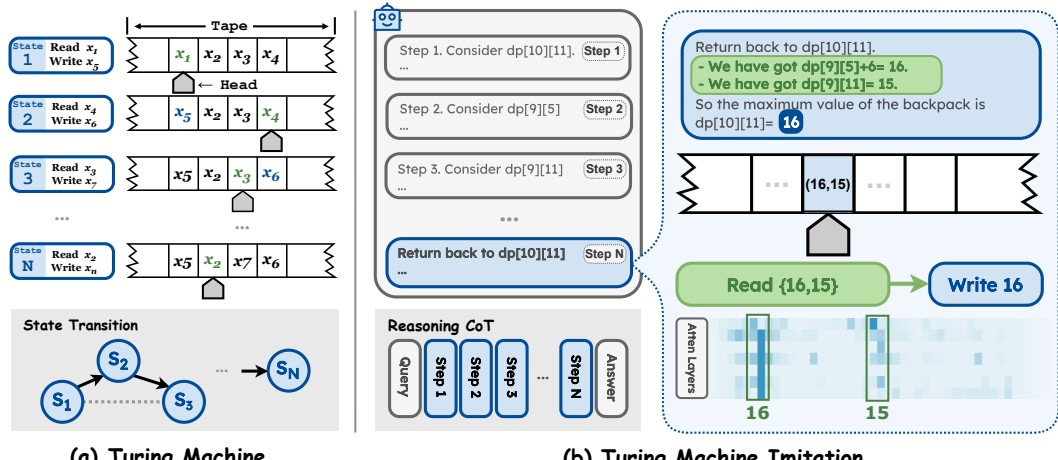

(a) Turing Machine (b) Turing Machine Imitation

Figure 1: Turing machine and its imitation in LLMs. **(a)** Illustration of a Turing machine performing algorithmic execution over a symbolic tape via sequential state transitions. **(b)** TAIL simulates Turing machine execution by linearly structuring CoT into atomic read-write steps. Attention maps reveal operand retrieval and memory update patterns analogous to symbolic computation.

such tasks as **Computable Problems**, which serve as the focus of investigation in this paper. Thus, the core of achieving length generalization lies in letting LLMs faithfully simulate the execution process of the corresponding programs within their CoT for each problem. In essence, the LLM acts like a Turing machine (Figure 1a), performing a sequence of fundamental operations on a memory tape, guided by finite states and logical transitions.

From this perspective, we propose **T**uring m**A**chine **I**mitation **L**earning (**TAIL**), which contains the three key structures in the synthesized CoT data that emulate three core properties of Turing machine execution: Linear Transition, Atomic State, and Memory Fetcher. First, similar to the Turing machine execution process, **Linear Transition** enforces a complete and linear arrangement of reasoning steps to eliminate potential shortcut learning. Second, TAIL decomposes the reasoning content into minimal units, termed **Atomic States** to reduce difficulty and further reduce shortcut learning, which essentially correspond to the states of a Turing Machine, including read, write, and logical control operations. Third, because LLMs can only append instead of modify in-place the tokens in their context due to their auto-regressive nature, the context of LLMs, which essentially serves as a memory, will keep growing as the reasoning continues. This poses difficulties for LLMs because of their attention mechanisms when they need to conduct elementary operations on operands that have long and dynamic distances among them. Therefore, TAIL further adopts a mechanism, termed **Memory Fetcher**, to read the necessary operand data and explicitly output them in the current step before conducting elementary operations.

To assess the universality and effectiveness of TAIL, we construct a challenging dataset spanning 18 tasks across 8 algorithms, substantially harder than those in prior length generalization studies. Fine-tuning Qwen2.5-7B (Yang et al., 2024) on this dataset yields high label accuracy across length ranges, with consistent gains on longer sequences, demonstrating strong length generalization over difficult samples. The model outperforms prior methods (Zhou et al., 2024; Lee et al., 2023; Shen et al., 2023; Zhou et al., 2023; Martínez et al., 2023; McLeish et al., 2024) and surpasses DeepSeek-R1 (Guo et al., 2025). Ablation studies show that removing any core module of TAIL severely degrades long-sequence performance. Notably, even minimalist CoT data containing only core modules without any thinking styles[1] maintains full effectiveness, confirming TAIL as the key data-driven enabler of length generalization. We also visualize the attention maps of the TAIL-fine-tuned model and observe that the attention during write operations focuses on fetched operands within the same state, resembling Turing machine behavior (Figure 1b).

---

[1]*Thinking styles* refers to the human-like linguistic expressions in CoT reasoning which are very common in existing large reasoning models, *i.e.*, the surface-level natural language narrative rather than the underlying reasoning mechanism.

## 2 PRELIMINARIES

### 2.1 LENGTH GENERALIZATION AND COMPUTABLE PROBLEMS

For large language models (LLMs), length generalization means that a model can process long input sequences, although it is only trained on short sequences. For example, a model trained on 10-30 digit addition can maintain strong performance on 30-50 digit addition tasks. Fundamentally, successful length generalization implies that the model has extracted a structural pattern from the training data. This pattern should be general and can scale adaptively with input length.

After a deeper look at the problem, incorporating insights from prior indirect conclusions (Delétang et al., 2022), we observe that many tasks can essentially be solved through discrete symbolic transformations governed by bounded algorithmic computational rules (Turing et al., 1936; Sipser, 1996; Arora & Barak, 2009; Boolos et al., 2002). For example, Parity can be solved through a simple enumeration procedure, while arithmetic addition can be handled by simulating the full digit-wise addition process, including carry propagation. We refer to such tasks as *Computable Problems*, whose commonality lies in being solvable by a well-defined, deterministic algorithmic procedure. Such algorithms inherently handle inputs of arbitrary length, which aligns with the goal of length generalization. Training LLMs to learn their step-by-step execution thus enables generalization across input lengths when solving computable problems.

### 2.2 TURING MACHINE

While all computable problems are solvable by algorithms, their structural diversity makes chain-of-thought (CoT) design impractical. Therefore, a more abstract and general framework is essential to unify the CoT paradigm for computable problems. Based on the Church-Turing thesis (Copeland, 1997), a Turing machine can solve any algorithmically computable problem, thereby providing a universal and higher-level framework for problem solving. In other words, the computational trace data of any computable problem can be constructed by simulating the execution of a Turing machine.

The formal definition of the Turing machine (Turing et al., 1936; Hopcroft et al., 2001) consists of an infinite-length tape, a read/write head, and a table containing a finite set of state transitions. It can be represented as a 7-tuple:

$$M = (Q, \Sigma, \Gamma, \delta, q_0, B, F), \tag{1}$$

where $Q$ is a finite set of states, $\delta$ is the transition function, $q_0 \in Q$ refers to the initial state (see Appendix A for full definitions). In any non-accepting state $q_s \in Q$, the head reads a symbol $a$ from the tape, overwrites it with a new symbol $b$, and moves the head to a new position, thereby transitioning to the next state $q_{s+1}$, which can be formally defined as:

$$\delta(q_s, a) = (q_{s+1}, b, D), \tag{2}$$

where the head moves one position in direction $D$. Thus, $\delta$ represents a complete state transition conflating two logically independent states, and a linear unfolding of states $q_0 \to q_1 \to \cdots \to q_n$ represents the complete process of Turing machine implementing the program. In order to align the reasoning process of LLMs with Turing Machine, the reasoning procedure can be unfolded into multi-step reasoning with the help of CoT. Each single reasoning step can be formalized as $x_i$ in CoT, deriving the current reasoning result based on the preceding reasoning steps $x_{<i}$. It is important to note that the granularity of $x_i$ is determined by the size of the reasoning step in the specific task. Typically, it corresponds to the prediction of multiple tokens, forming an intermediate reasoning outcome at each stage. $x_0$ represents the query and thus the complete reasoning path (CoT) can be expressed as $x_0 \to x_1 \to \cdots \to x_n$. In line with the Turing machine, each reasoning step $x$ corresponds to a Turing Machine state $q$ in Eq.(2), which includes reading an input symbol $a$. The entire CoT is formed by a linear composition of such steps, analogous to the full unrolling of the Turing Machines state transitions $\delta$ from $q_1$ to $q_n$.

Previous work (Li et al., 2024) has theoretically shown that Transformers can achieve Turing Completeness given sufficiently long CoT, but has not provided concrete guidelines for constructing such CoT sequences in a wide range of tasks. Based on our analysis, multi-step reasoning CoT can be structurally aligned with the computation process of a Turing machine. This leads us to hypothesize that, by endowing the reasoning process of LLMs with key properties of a Turing machine, the model can effectively simulate algorithmic execution and achieve length generalization.

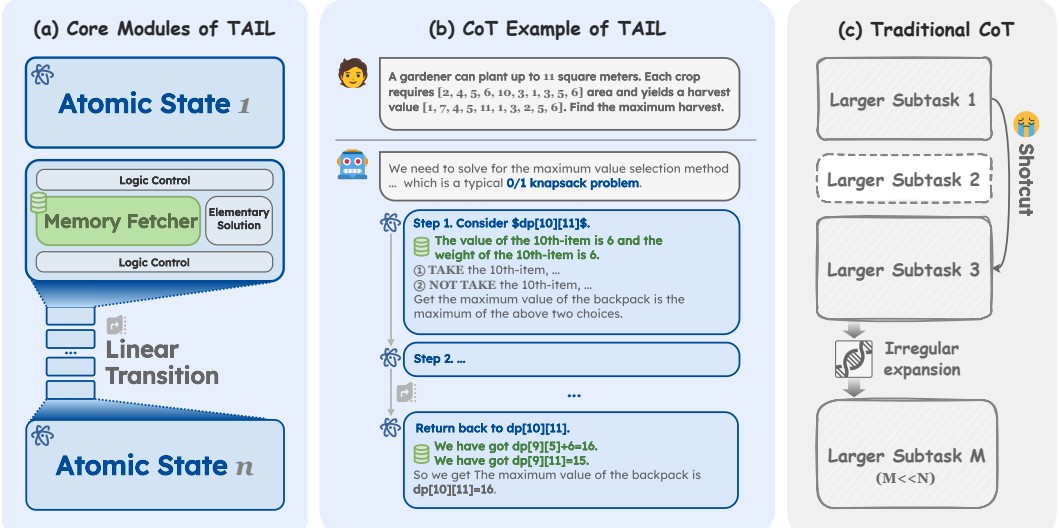

Figure 2: An overview of TAIL. **(a) Core Modules of TAIL** imitate a Turing Machine, containing a Linear Transition of Atomic State with Memory Fetcher of previous reasoning results. **(b) CoT generated by TAIL**: the solution to a 0/1 knapsack problem using a dynamic programming algorithm. **(c) Traditional CoT** consists of oversized subtasks, shortcut learning, and irregular expansion.

## 3 TURING MACHINE IMITATION LEARNING

Based on the preceding analysis, this paper proposes **Turing mAchine Imitation Learning (TAIL)** to align the Chain-of-thought (CoT) of large language models (LLMs) to simulate the execution of a Turing machine for achieving universal and effective length generalization. TAIL imitates key properties of a Turing Machine (Figure 2), comprising three core modules spanning macro to micro levels: Linear Transition (Section 3.1), Atomic State (Section 3.2) and Memory Fetcher (Section 3.3).

### 3.1 LINEAR TRANSITION

According to the RASP-Generalization Conjecture (Zhou et al., 2023), Transformer-based LLMs struggle with problems that involve intricate control structures, such as loops. This suggests the need to transform these structures into simpler forms that align better with the capabilities of the model. In particular, complex reasoning structures (like trees and graphs) can be linearly unrolled and traversed to enable complete and non-redundant execution of all reasoning steps, thereby preventing shortcuts in the reasoning process. Similarly, in a Turing machine, the execution of a complete program corresponds to a linear unfolding of states $q_1 \rightarrow q_2 \rightarrow \cdots \rightarrow q_n$ as shown in Eq.(2), where even control structures such as loops can be flattened into a sequential process. To align with this characteristic, we introduce Linear Transition, which describes from a macro-level perspective how individual reasoning steps are composed into a linear and orderly structure within the overall reasoning process, and collectively form the CoT.

### 3.2 ATOMIC STATE

Although Linear Transition defines the overall structure of CoT reasoning as a linear sequence of reasoning steps, it does not impose constraints on the size of each step. Overly large reasoning steps not only increase the difficulty of learning for the model but also risk introducing shortcuts within a single step. Therefore, we attempt to constrain the size of a reasoning step by enforcing a standardized internal structure. Inspired by the Turing machine, each state encompasses a sequence of simple operations: *reading* data from the tape, *writing* new data, and *transitioning* to the next state. Following this principle, we define Atomic State consisting of operand retrieval (realized via Memory Fetcher, detailed in Section 3.3), the elementary solution produced within the reasoning step, and a set of logical control statements, as shown in Figure 2(a). Meanwhile, following the

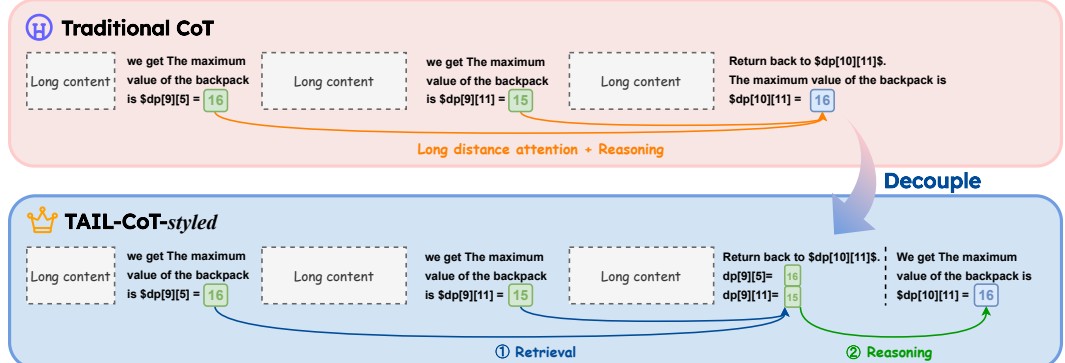

Figure 3: Comparison of traditional CoT and TAIL's *Memory Fetcher*, which decouples long-distance attention and the reasoning process.

RASP-L hypothesis[2], we argue that each Atomic State should adhere to the principles of realizability and simplicity. Specifically, since we use Python programs to synthesize CoT, we define an Atomic State as a single algorithmic step in the program without internal loops.

### 3.3 MEMORY FETCHER

In a Turing machine, every state reads data from the tape and replaces it with processed result. However, auto-regressive models (*e.g.*, Transformer) can only extend the token sequence by appending new tokens instead of in-place token modification. So the action of data reading is typically achieved by constructing attention mechanisms over previous tokens. As reasoning progresses, the sequence grows longer, requiring the model to retrieval over increasingly distant and dynamically shifting tokens. Furthermore, simultaneously performing data retrieval and generating the elementary solution at the same time increases the learning difficulty for the model. To address this, we propose Memory Fetcher to decouple these two operations by: (1) *first* explicitly outputting all relevant operands at the beginning of every Atomic State, (2) *then* performing reasoning and outputting local results. As shown in an example in Figure 3, Memory Fetcher changes the attention structure by localizing operands and improves reasoning accuracy, which has been theoretically proved by recent work (Wang et al., 2025). Figure H1 compares the attention structures with and without Memory Fetcher. It is obvious that Memory Fetcher enables precise localization of relevant operands during reasoning through prominent local attention. See more details in Appendix H.

## 4 EXPERIMENT

### 4.1 DATASET SYNTHESIS

**Task Selection.** This work focuses on length generalization in hard samples rather than unlimited extension in simple tasks. So we synthesize a set of *challenging* tasks based on 8 classic algorithmic paradigms in computable problems to verify the effectiveness of TAIL. As shown in Table C1, the dataset comprises 18 tasks, including previously studied problems such as addition, but with randomized digit lengths and decimal places to increase difficulty. Each task has a high degree of diversity in query narratives, some of which incorporate real-world problems (*e.g.*, Diophantine Equation, 0-1 Knapsack, etc).

**Synthesis Approach.** We employed supervised fine-tuning (SFT) with synthetic data to internalize the model's ability to generate Chain-of-Thought (CoT) with TAIL's core modules. Figure D1 illustrates the data synthesis process of TAIL. We claim that TAIL is task *universal*[3] because for each task belonging to a specific algorithm, it's feasible to construct a Python program and add

---

[2]We do not strictly follow RASP-L to constrain each reasoning step, but use it to indicate problems directly solvable by Transformers. This relaxed view shows strong length generalization in our experiments.

[3]Previous methods like Index Hint or Reversed Format, due to their structural specificity, cannot be effectively constructed for tasks beyond simple bit-matching operations.

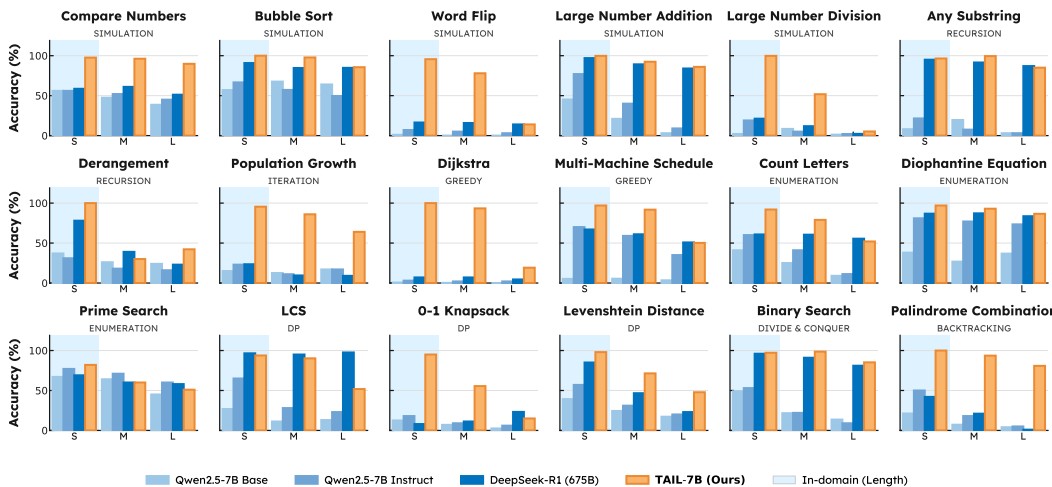

Figure 4: Length generalization performance of Qwen2.5-7B finetuned with TAIL-CoT-*style* across all 18 tasks, in comparison with Qwen2.5-7B (base model), Qwen2.5-7B Instruct and DeepSeek-R1.

string append statements to assemble CoT. When the program runs, the resulting CoT reflects the complete program execution flow. We implement the injection of three core modules in CoT through the following methods: (1) Treating each algorithmic step as an Atomic State, especially each time entering a loop. (2) Unfolding the algorithmic process sequentially as Linear Transition, achieved by using programs to synthesize CoT itself. (3) Explicitly outputting all relevant operands of current algorithm step as Memory Fetcher in CoT. During the generation process, we performed strict de-duplication and ensured that none of the data in the evaluation set was included in the training set. For training data, we first validated the sufficiency of the TAIL architecture by synthesizing **TAIL-CoT** that only includes three core modules in a format similar to that shown in Figure D3. Then we enrich TAIL-CoT into **TAIL-CoT-*styled*** using natural language (see Figure D4) and verify length generalization ability on all 18 tasks, as shown in Figure 4. See Appendix D for more details.

**Dataset Size.** To better facilitate training and evaluation of length generalization, we defined three length ranges for each task: Short (**S**), Medium (**M**) and Long (**L**). For comprehensive training, we synthesized 100,000 training samples and 500 evaluation samples for each length range, resulting in 1,500 evaluation samples per task. The validation of length generalization refers to whether a model trained on the S-range training set can avoid sharp performance degradation on the M- and L-range evaluation sets. The length ranges of each task are detailed in Table F1.

## 4.2 EXPERIMENTAL SETTINGS

**Metrics.** Previous work (Saparov & He, 2022) has demonstrated experimentally that *label accuracy* is well suited to measure reasoning capability of LLMs. We use pass@1 label accuracy under the zero-shot setting and use greedy decoding to evaluate.

**Training.** We fine-tuned Qwen2.5-7B with training 2 epochs for most tasks and more epochs for a few more challenging ones with a global batch size of 1024. The initial learning rate was 1e-5, decaying to 7e-7, with a weight decay of 0.1.

**Evaluation.** To facilitate a more efficient evaluation procedure, we follow a dual-model framework. First, a small 1.5B specialized model extracts the answers. Then, Qwen2.5-72B-Instruct performs evaluations, outputting \boxed{YES} or \boxed{NO} to represent the evaluation result.

## 4.3 PERFORMANCE

**Overall Performance.** Due to its greater readability, we synthesized TAIL-CoT-*styled* (as shown in Figure D4) on all 18 tasks across 8 classes of algorithms, and fine-tuned Qwen2.5-7B. As shown in Figure 4, We observe **length generalization on most difficult tasks**, where there was no sharp performance degradation on out-of-domain length sequences. Several tasks like *Compare Numbers*,

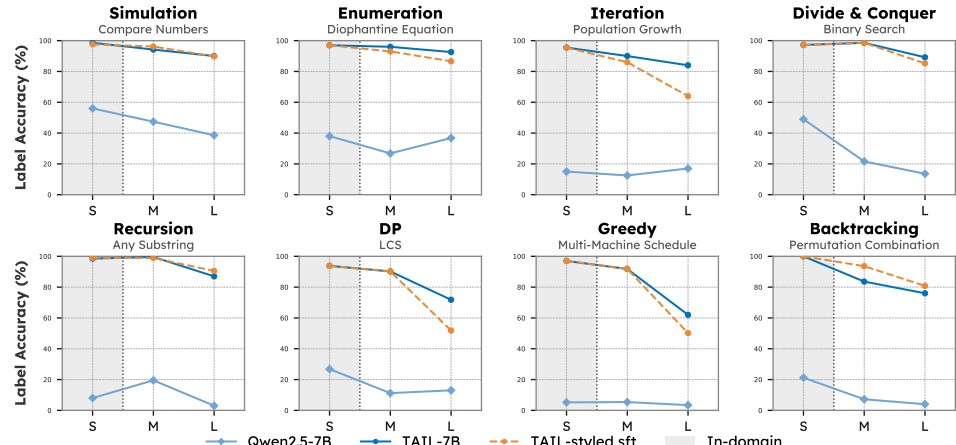

Figure 5: Comparison of fine-tuned Qwen2.5-7B with TAIL core module and the base model. For each algorithm, we select a representative task. After fine-tuning, model demonstrates length generalization on sequences that are 5 to 10 times longer than those in training.

| Model | Simulation | | Enumeration | | Iteration | | Divide & Conquer | |
|---|---|---|---|---|---|---|---|---|
| | M | L | M | L | M | L | M | L |
| Qwen2.5-7B Base | 47.4 | 38.6 | 26.8 | 36.8 | 12.4 | 17.0 | 21.6 | 13.6 |
| w/o Atomic State | 82.6 | 73.6 | 68.2 | 69.0 | 77.2 | 61.2 | 86.2 | 71.0 |
| w/o Linear Transition | 80.0 | 75.4 | 63.6 | 58.8 | 76.2 | 54.0 | 85.0 | 75.6 |
| w/o Memory Fetcher | 90.2 | 88.0 | 64.2 | 63.6 | 73.8 | 67.6 | 92.4 | 88.2 |
| TAIL | **94.2** | **90.0** | **96.0** | **92.6** | **90.0** | **84.4** | **98.6** | **89.2** |

| Model | Recursion | | DP | | Greedy | | Backtracking | |
|---|---|---|---|---|---|---|---|---|
| | M | L | M | L | M | L | M | L |
| Qwen2.5-7B Base | 19.4 | 3.0 | 11.2 | 13.0 | 5.4 | 3.4 | 7.2 | 4.0 |
| w/o Atomic State | 52.2 | 32.0 | 77.4 | 61.0 | 39.0 | 16.0 | 75.8 | 61.4 |
| w/o Linear Transition | 43.0 | 30.8 | 77.4 | 74.2 | 20.6 | 11.2 | 79.0 | 62.8 |
| w/o Memory Fetcher | 87.2 | 84.8 | 80.8 | 74.8 | 45.6 | 30.8 | 80.0 | 69.2 |
| TAIL | **99.6** | **87.0** | **90.2** | **71.8** | **91.8** | **62** | **83.6** | **76** |

Table 1: Ablation study in core modules of TAIL. For each algorithm, we select a representative task and evaluate pass@1 accuracy (%) only on sequences that exceed the training length. Results show that the absence of any core module leads to a sharp degradation in length generalization performance.

*Bubble Sort* and *Any Substring* reach near saturation in out-of-domain sequences. Moreover, TAIL also outperformed Qwen2.5-7B (base model), Qwen2.5-7B Instruct (fine-tuning on a large amount of traditional non-TAIL-CoT data), and DeepSeek-R1 671B (representative open-source reasoning model) in both label accuracy and length generalization abilities. Compared with reasoning models (*i.e.*, DeepSeek-R1), we conclude that the huge leap in performance lies in their **different underlying mechanisms**. Reasoning models often try many approaches but only scratch the surface and exploit shortcuts to bypass the structured reasoning process, instead of delving into a step-by-step approach (see more details in section 4.5). Regarding concerns about CoT length, experiments show that TAIL-CoT achieves a significant performance leap with **slightly less token usage** compared to inference models, as shown in Appendix J.

| Method | S | M | L |
|---|---|---|---|
| Index Hint | 57.0 | 34.5 | 24.0 |
| Reversed Format | 39.5 | 35.5 | 35.0 |
| **TAIL (Ours)** | **97.0** | **92.5** | **86.5** |

Table 2: Pass@1 (%) on large-number addition, compared to previous data-driven approaches.

**Comparison with Previous Works.** Since previous methods (Index Hint (Zhou et al., 2024; Lee et al., 2023; Shen et al., 2023; Zhou et al., 2023; Martínez et al., 2023; McLeish et al., 2024) and Reversed Format (Zhou et al., 2023; 2024)) have proven effective on limited problems such as large number operations, we choose *Large Number Addition* of *Simulation* algorithm as a com-

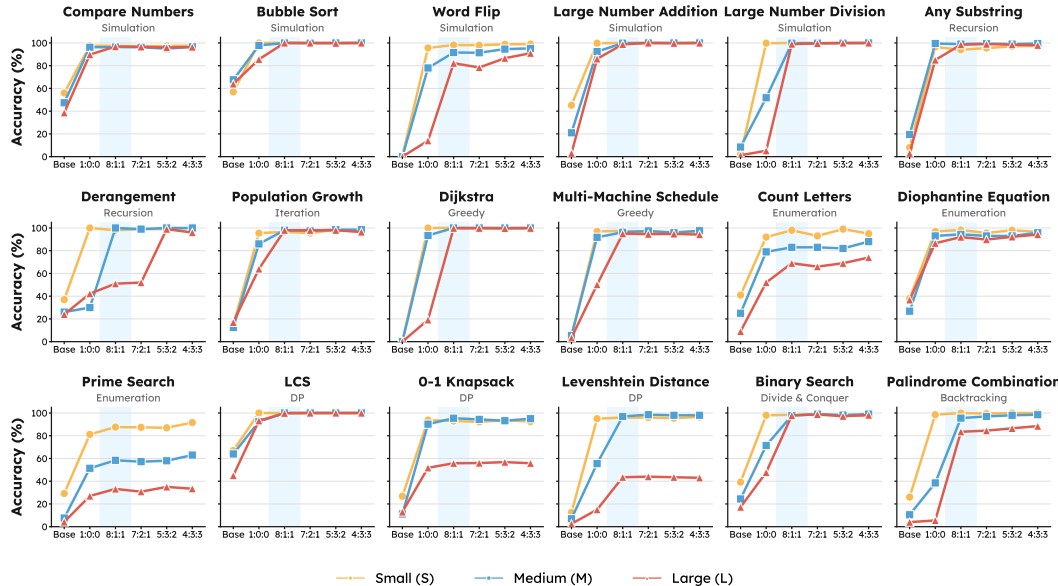

Figure 6: Experiments on the proportion of training data of different lengths. We find that adding a small amount of long data to most unsaturated tasks achieves fast performance gains.

mon task for comparison. Unlike prior work using fixed-length integers, our setup samples two operands with random lengths and optional **decimal points**, greatly expanding the state space. We followed the method in previous works to construct the same amount of training data (see details in Appendix G), and fine-tuned Qwen2.5-7B separately. As shown in Table 2, models trained with In-dex Hint and Reversed Format under-perform TAIL by a large margin, highlighting the inadequacy of prior methods in addressing the challenges of length generalization in difficult tasks.

**Length Generalization Activation.** In previous experiments, we only trained on S-range data to evaluate the generalization performance on longer sequences (M, L). For tasks that haven't achieved saturation, we gradually introduce longer examples into the training set and analyze the optimal proportion for effective length generalization at minimal token cost. We explored five training con-figurations of $<$S,M,L$>$ data while keeping the total number of samples constant: $<1:0:0>$ (the previous method using only short sequences), $<8:1:1>$, $<7:2:1>$, $<5:3:2>$, and $<4:3:3>$. As shown in Figure 6, for almost all tasks, even a small addition of longer sequence data (*i.e.*, at $<8:1:1>$) led to a rapid saturation in long-sequence reasoning, a phenomenon we refer to as *length generalization activation*. This observation is quite different from the "balanced length" conclusion of training data in previous works (Lee et al., 2023), indicating that TAIL has the potential to expand to much longer sequences at a lower cost in the future. See more details in Appendix E.

## 4.4 ABLATION STUDY

**Key module ablation.** To assess the necessity of each core module, we ablate them individually and examine the performance drop. As shown in Table 1, removing any module leads to a notable de-cline in length generalization. Importantly, the impact varies by task: for example, Memory Fetcher is critical for Population Growth (iteration-based), but less so for Compare Numbers (simulation-based). We attribute this variation to differences in task structure. Tasks like Compare Numbers involve only local transitions and weak long-range dependencies, making Memory Fetcher less es-sential. In contrast, recursion-heavy tasks benefit significantly from Linear Transition. Most tasks are also sensitive to scale, highlighting the general need for Atomic State decomposition. Overall, TAIL integrates all three modules synergistically to support diverse reasoning structures.

**Thinking style ablation.** To investigate the influence of different CoT styles on performance, we conducted fine-tuning experiments using both standard TAIL-CoT and TAIL-CoT-*styled* data. As illustrated in Figure 5, the results indicate that the choice of CoT style has minimal impact on the final performance. This suggests that for the length generalization task, the specific style of CoT

is not a critical factor. Instead, key modules of TAIL appears to play a more significant role in determining the overall performance.

**Attention visualization of Memory Fetcher.** As shown in Figure H1, when Memory Fetcher is present, we observe strong and focused attention on the corresponding tokens (highlighted in selected Transformer layers). In contrast, the attention patterns become sparse and disorganized without Memory Fetcher, showing insufficient focus on the operands. See more details in Appendix H.

### 4.5 COMPARISON WITH REASONING MODELS

It seems that both TAIL-CoT(-*styled*) and reasoning models (*i.e.*, DeepSeek-R1) can improve performance by extending the CoT length, but underlying principles are quite different. Reasoning models aim to expand the search space by prolonging the reasoning trajectory, encouraging **broad method exploration** instead of delving into the problem step by step (as shown in Section L.1). In contrast, TAIL focuses on **controllable** and **structured** reasoning chains that support stable generalization to longer sequences (as shown in Section L.2). See Appendix L for more details.

Despite appearances suggesting that the linear expansion of small reasoning steps with explicit operands output make TAIL-CoT(-*styled*) longer than traditional CoT, experimental results show that TAIL-CoT can achieve significantly higher accuracy at comparable CoT length to DeepSeek-R1 (as shown in Table J1). It demonstrates that TAIL's core modules are essential for promoting length generalization. See Appendix J for more details.

To rigorously evaluate the difference, we also fine-tuned Qwen2.5-7B using an equal amount of correct data distilled by DeepSeek-R1. As shown in Table K1, R1-Distilled-Qwen2.5-7B exhibits lower accuracy and generalization ability. See Appendix K for more details.

## 5 RELATED WORK

**Length Generalization.** Large language models (LLMs) often struggle to process inputs longer than those seen during training, a limitation referred to as length generalization (Dubois et al., 2019; Newman et al., 2020; Saparov & He, 2022; Anil et al., 2022). Previous works primarily focus on model architecture enhancements and data-driven approaches to improve length generalization. However, architectural enhancements modify the components (*e.g.*, forward mechanism (Fan et al., 2024), attention mechanisms (Duan et al., 2023), position encodings (Ruoss et al., 2023; Li et al., 2023; Kazemnejad et al., 2023) and external queries (Giannou et al., 2023)) of Transformers for specific tasks and further adaptation to be applicable to prevailing LLMs. Data-driven approaches construct specific chain-of-thought (CoT) structures for training, such as digit-order reversal (Zhou et al., 2024; Lee et al., 2023; Shen et al., 2023; Zhou et al., 2023; Martínez et al., 2023; McLeish et al., 2024), sequence padding (Jelassi et al., 2023), and index hints (Zhou et al., 2023; 2024), which are task-specific and lack universality. Our TAIL focuses on universal data-driven approaches, exploring a more general and effective CoT structure, and directly adopting mainstream LLMs (Bai et al., 2023; Yang et al., 2024; Touvron et al., 2023; Liu et al., 2024; Team, 2023; Cai et al., 2024; Bai et al., 2025) for fine-tuning, without modifying any components of the pretrained model. Similar to ours, recent work (Hou et al., 2024) adopts a Turing-like step-by-step tape update, but it is limited to specific positional encodings and data settings, and lacks verification across diverse tasks.

**Structured Chain-of-Thought Construction.** Structured thinking demonstrably enhances the reasoning capabilities of LLMs (Wei et al., 2022). Prior research explored various recognition heuristics within CoT paradigm, aiming to imbue LLMs with more human-like thoughts (Suzgun & Kalai, 2024; Zou et al., 2023; Zheng et al., 2023). Concurrently, investigations into diverse structured data formats, including linear chains (Wei et al., 2022), hierarchical trees (Yao et al., 2023a), interconnected graphs (Besta et al., 2024), and dynamically adapting structures (Pandey et al., 2025), which enable LLMs to search easily and improve the complex problem-solving performance. In this paper, we introduce a novel approach to synthesizing structured CoT data by drawing inspiration from a Turing machine, which can handle inputs of arbitrary length. This emulation offers a theoretically powerful advantage: length-generalizability, enabling the model to tackle problems of varying complexity, and broad applicability to the entire domain of *computable* problems. Notably, the search capability and different graph structures (Besta et al., 2024; Pandey et al., 2025; Yao et al.,

2023a) and their targeted tasks can all be taken as instances of a Turing machine solving computable problems.

**Eliciting Chain-of-Thought.** Efforts to guide LLMs in generating CoT rationales with specific, desired structures primarily involve prompting, supervised fine-tuning (SFT), and reinforcement learning (RL). Prompting-based methods, encompassing static example-based approaches Shao et al. (2023); Zhou et al. (2022); Wang et al. (2023) and adaptive generation strategies Shum et al. (2023); Yao et al. (2023b); Wang & Zhou (2024), guide the model through carefully crafted prompts. However, these methods do not fundamentally enhance LLMs' inherent reasoning capabilities Wang et al. (2024); Rueda et al. (2025). SFT methods Yu et al. (2025); Trung et al. (2024); Kim et al. (2023); Zhang et al. (2024) emphasize guiding single or multiple LLMs to synthesize training data with specific CoT structures, allowing the model to learn and imitate the target reasoning behavior. RL-based methods Xiang et al. (2025); Tang et al. (2025); Shen et al. (2025) allow the model to generate its own CoT reasoning traces and optimize via a reward function, where the resulting structures emerge during training and remain partly uncontrollable Xu (2025); Liu et al. (2025). Despite the progress of SFT and RL, a persistent challenge remains to mitigate shortcuts and logical omissions during CoT reasoning, especially when relying on model-generated rollouts Lightman et al. (2023); Zhang et al. (2024); Wu et al. (2024); Yu et al. (2025). To address this, we explore using Python programs to synthesize SFT CoT data to guarantee that the structure of CoT strictly mimics the execution process of a Turing Machine and reduce the possibility of learning shortcuts.

## 6 LIMITATIONS

**Challenges in compositional generalization.** Although TAIL improves the length generalization performance on each single task, the training of one task does not significantly improve the performance of other tasks under the same algorithm (as shown in Appendix I). In future work, we will take individual tasks as the entry point, to explore more diverse data composition strategies, with the goal of achieving compositional generalization.

**Gap with close-source models.** In the experiment, we found that open-source models such as Qwen2.5 and DeepSeek-R1 series did not perform well on our tasks, but several closed-source models (*e.g.*, O4-mini) were able to solve these problems well. While we acknowledge the strong performance of closed-source models, our focus is on bridging this gap solely through supervised fine-tuning of open-source models with TAIL data.

**Challenges in modeling non-deterministic algorithmic tasks.** The scope of this work is limited to computable problems, and the core idea of simulating a Turing Machine is based on this assumption. However, for non-deterministic problems or open-ended reasoning, we cannot directly model an algorithm to solve it, which is a problem that TAIL cannot currently solve. We will actively explore ways to break through the boundaries of computable and fuzzy problems with structured CoT in future work.

## 7 CONCLUSION

We introduced Turing mAchine Imitation Learning (TAIL), a data-driven framework that instantiates three core modules (*i.e.*, Linear Transition, Atomic State, and Memory Fetcher) to align CoT structure with program execution and thereby promote universal and effective length generalization. Across 8 algorithm classes and 18 tasks, fine-tuning on TAIL-synthesized data yields strong length generalization on out-of-distribution sequence lengths, with consistent gains on difficult cases and performance that surpasses DeepSeek-R1. Unlike reasoning models that expand trajectories to explore many heuristics, TAIL enforces a controllable and step-by-step execution, which supports stable extrapolation to inputs of arbitrary length.

ACKNOWLEDGMENTS

We thank the anonymous reviewers and area chair for their helpful comments. This project is supported by the Shanghai Artificial Intelligence Laboratory. The authors would like to thank Xun Zhao for her valuable suggestions and comments.

STATEMENT OF LLMS USAGE

In this paper, we used LLMs to aid and polish writing. We did not use LLMs for retrieval, discovery or research ideation.

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

## A    FORMAL DEFINITION OF TURING MACHINE

A Turing machine (Turing et al., 1936; Hopcroft et al., 2001) can be formally defined as a seven-tuple:

$$M = (Q, \Sigma, \Gamma, \delta, q_0, B, F), \tag{A1}$$

where $Q$ is a finite set of states, $\Sigma$ is a non-empty finite input alphabet, $\Gamma$ denotes the set of tape symbols, $\delta$ is the transition function, $q_0 \in Q$ refers to the initial state, $B \in \Gamma - \Sigma$ is the blank symbol, and $F$ denotes the set of final states.

## B    THE UNDERLYING MOTIVATION OF TAIL

> **TL;DR.** To improve length generalization, TAIL resolves two fundamental limitations of traditional CoT: **shortcut reasoning** and **sparse long-distance attention**. The *Atomic State* and *Linear Transition* address shortcut behavior at the micro and macro levels respectively, while the *Memory Fetcher* mitigates the sparse-attention issue.

### B.1    SHORTCUT REASONING

Previous works Anil et al. (2022); Saparov et al. (2024) show that when SFT'd on non-sequential CoT, LLMs prefer to take shortcuts during inference, and subsequently fail to generalize to longer sequences. From a micro perspective, shortcuts appear as **overly large single-step reasoning**, which means the model jumps to substantial sub-results instead of reasoning step by step Anil et al. (2022); Saparov et al. (2024); From a macro perspective, we also observe shortcut behavior as **step-skipping** and **frequent switching between reasoning strategies** (see Appendix L.1 for CoT example of DeepSeek-R1).

- **Atomic State** focuses on the micro level of each reasoning step. Guided by the RASP-L theory Zhou et al. (2023), it produces length-generalizable steps of appropriate granularity (*i.e.*, steps without internal loops).
- **Linear Transition** focuses on the macro structure of connecting reasoning steps. It linearly unfolds the entire reasoning process step by step, and the tight logical linkage between Atomic States prevents step-skipping.

### B.2    SPARSE LONG-DISTANCE ATTENTION

Previous work Wang et al. (2025) shows that long-sequence reasoning leads to severe attention sparsity in later steps. As illustrated in Figure H1 (lower part), traditional CoT attends to distant past states while performing the current reasoning action, resulting in sparse long-distance attention. Memory Fetcher decouples long-range attention construction from action execution: (1) First build long-range attention via a simple copy operation, copying the relevant operands to the end of the sequence; (2) then performs the reasoning action.

As shown in Fig H1 (upper part), introducing Memory Fetcher significantly improves attention distribution during reasoning actions. More attention is focused on nearby relevant operands, improving single-step correctness, and thereby stabilizing long-sequence reasoning.

# C   TASK INTRODUCTION

The dataset consists of purely synthetic data, covering 8 major algorithms and 18 tasks, as shown in Table C1. Most tasks have approximately 100,000 training samples and 500 test samples (a small subset of tasks, which are more difficult to construct, retain 20,000 training samples and 200 test samples). All test queries have been verified and are not included in the training set.

| Algorithm | Task Name | Task Content |
|---|---|---|
| Simulation | Large Number Addition | $x_1 + x_2$ * $(\text{len}(x_1) = n, \text{len}(x_2) = m)$ |
| | Large Number Division | $x_1 \div x_2$ * $(\text{len}(x_1) = n, \text{len}(x_2) = m)$ |
| | Bubble Sort | Bubble sort list of $n$ non-repeat numbers |
| | Word Flip | Flip a sentence containing $n$ letters |
| | Compare Numbers | Compare $x_1$ and $x_2$ * $(\text{len}(x_1) = n, \text{len}(x_2) = m)$ |
| Recursion | Any Substring | Find all substrings of given string with length $n$ |
| | Derangement | Derangement count for $n$ elements |
| Iteration | Population Growth | Calculate total pairs after $n$ units, starting reproduction at $x$-th unit $(x < n)$ with initial $y$ pairs, $z$ pairs produced per unit $(y, z \in \mathbb{N}^+)$ |
| Greedy | Dijkstra | Shortest path values in graph with $n$ vertices |
| | Multi-Machine Schedule | Maximum benefit of $n$ tasks on $x$ queues $(x < n)$ |
| Enumeration | Count Letters | Count letters in sentence of length $n$ |
| | Diophantine Equation | Find integer solutions to $x_1 a + x_2 b = n$ $(a, b \geq 0)$ |
| | Prime Search | All prime numbers in the interval $[n, m]$ $(n < m)$ |
| DP | 0-1 Knapsack | Maximum benefit of $n$-element 0-1 knapsack |
| | LCS | LCS of string $X_1$ and $X_2$ $(\text{len}(X_1) = n, \text{len}(X_2) = m)$ |
| | Levenshtein Distance | Minimum operations converting string $X_1$ to $X_2$ |
| Divide & Conquer | Binary Search | Binary search index in list of $n$ increasing numbers |
| Backtracking | Permutation Combination | Number of combinations in $n$-element list (step by step) |

Table C1: Dataset synthesised under instruction of TAIL, containing 8 algorithms and 18 tasks. $n$ and $m$ represent length in a given range $G \in \{S, M, L\}$. * indicates that a decimal point can be inserted in any bit of the operand in specific task.

# D  DATA SYNTHESIS AND COT EXAMPLES

Figure D1 illustrates our data synthesis process. Specifically, for each task, we manually write a Python program that can accept any input under that task. Because each task is assigned to an algorithm, writing this program is very convenient and well-reasoned. We then add string concatenation statements to the program to link the reasoning process and form a complete Chain-of-Thought (CoT). Since CoTs are generated as the program executes, they completely follow the program's execution process, which is the core idea behind TAIL. As the synthetic CoTs strictly follows the running process of Python programs, they will *exhaustively explore all possible solutions*.

During this process, we can output two types of CoT: (1) **TAIL-CoT** contains only the TAIL core module, without any other verbiage and is more symbolic. (2) **TAIL-CoT-*styled*** adds more style statements and is more human-readable and interpretable, which adds more cohesive and planning statements.

In experiments, we trained TAIL-CoT-*styled* on 18 tasks across all 8 algorithm classes and verified its strong length generalization performance, as shown in Figure 4. We then verified that removing all explicit style statements (TAIL-CoT) did not lead to a performance degradation, as shown in Figure 5, demonstrating that length generalization is the core module of TAIL, not the style statement.

Take *Binary Search* task in *Divide & Conquer* algorithm as an example. Figure D2 is the query as direct input to LLMs. For each task, we constructed more than 20 query templates to simulate the diversity. Figure D3 is an example of TAIL-CoT and Figure D4 is an example of TAIL-CoT-*styled*.

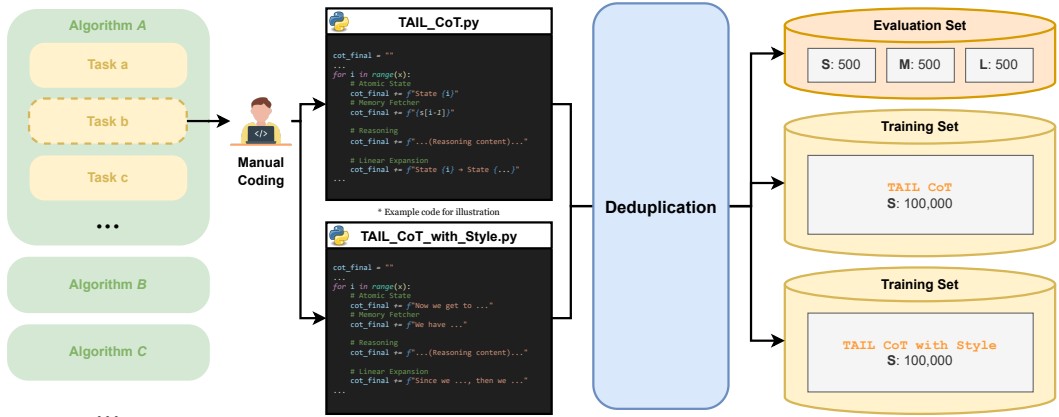

Figure D1: Overall pipeline of data synthesis. We manually code task-specific python programs and use them to massively generate chain-of-thought (CoT) data, either in plain form (TAIL-CoT) or with stylistic variations (TAIL-CoT-*styled*). After deduplication, we construct large-scale training sets and balanced evaluation sets for subsequent experiments. (**S** = Short sequence data, **M** = Medium sequence data, **L** = Long sequence data)

---

**Query: (Example 1)**

Determine the index of $-5259$ in the sorted list $[-5957, -5259, -4195, -2263, 1289, 3514, 3632, 4284, 5991, 6578, 7333]$ with binary search (start from $0$). Other search methods are not allowed.

**Query: (Example 2)**

Find the index of $-5259$ in the sorted list $-5957, -5259, -4195, -2263, 1289, 3514, 3632, 4284, 5991, 6578, 7333$, starting from $0$. For teaching purposes, you must use binary search and show the process step by step.

**Query: (Example 3)**

Provide the 0-based binary search index for $-5259$ in $[-5957, -5259, -4195, -2263, 1289, 3514, 3632, 4284, 5991, 6578, 7333]$.

---

Figure D2: Example queries belonging to the Binary Search task (Divide & Conquer algorithm).

---

**Minimalist reasoning chain of TAIL:**

<Atomic State> $\{0,10\}$
<Memory Fetcher> $[(s_0 = -5957), (s_1 = -5259), (s_2 = -4195), (s_3 = -2263), (s_4 = 1289), (s_5 = 3514), (s_6 = 3632), (s_7 = 4284), (s_8 = 5991), (s_9 = 6578), (s_{10} = 7333)]$
<Memory Fetcher> Find $= -5259$
Mid $= 5, s_5 = 3514 >$ Find
<Linear Transition> $\{0,10\} \rightarrow \{0,4\}$

<Atomic State> $\{0,4\}$
<Memory Fetcher> $[(s_0 = -5957), (s_1 = -5259), (s_2 = -4195), (s_3 = -2263), (s_4 = 1289)]$
<Memory Fetcher> Find $= -5259$.
Mid $= 2, s_2 = -4195 >$ Find
<Linear Transition> $\{0,4\} \rightarrow \{0,1\}$

<Atomic State> $\{0,1\}$
<Memory Fetcher> $[(s_0 = -5957), (s_1 = -5259)]$
<Memory Fetcher> Find $= -5259$
Mid $= 0, s_0 = -5957 <$ Find
<Linear Transition> $\{1,1\} \rightarrow \{1,1\}$.

<Atomic State> $\{1,1\}$
<Memory Fetcher> $[(s_1 = -5259)]$
<Memory Fetcher> Find $= -5259$
Mid $= 1, s_1 = -5259 =$ Find
<Linear Transition> $\{1,1\} \rightarrow F$

Final Answer: Index $\boxed{1}$.

---

Figure D3: Example of TAIL-CoT, with only core modules aligned with Turing machines. <TAG> is for demonstration purposes only and does not exist in the actual CoT content.

**User-friendly CoT of TAIL:**

Let's perform the binary search step by step.

1. The currently processed interval is [-5957, -5259, -4195, -2263, 1289, 3514, 3632, 4284, 5991, 6578, 7333].
First we map each number to its index:
- Original index 0: $-5957$
- Original index 1: $-5259$
- Original index 2: $-4195$
- Original index 3: $-2263$
- Original index 4: 1289
- Original index 5: 3514
- Original index 6: 3632
- Original index 7: 4284
- Original index 8: 5991
- Original index 9: 6578
- Original index 10: 7333
The left boundary index is 0 and the right boundary index is 10.
Median index $\lfloor (0+10)/2 \rfloor = 5$, the median number is 3514.
The target number $-5259$ is less than the median number 3514, continue these operations on the left subinterval $[-5957, -5259, -4195, -2263, 1289]$.

2. The currently processed interval is $[-5957, -5259, -4195, -2263, 1289]$.
First we map each number to its index:
- Original index 0: $-5957$
- Original index 1: $-5259$
- Original index 2: $-4195$
- Original index 3: $-2263$
- Original index 4: 1289
The left boundary index is 0 and the right boundary index is 4.
Median index $\lfloor 0+4 \rfloor = 2$, the median number is $-4195$.
The target number $-5259$ is less than the median number $-4195$, continue these operations on the left subinterval $[-5957, -5259]$.

3. The currently processed interval is $[-5957, -5259]$.
First we map each number to its index:
- Original index 0: $-5957$
- Original index 1: $-5259$
The left boundary index is 0 and the right boundary index is 1.
Median index $\lfloor 0+1 \rfloor = 0$, the median number is $-5957$.
The target number $-5259$ is greater than the median number $-5957$, continue these operations on the right subinterval $[-5259]$.

4. The currently processed interval is $[-5259]$.
First we map each number to its index:
- Original index 1: $-5259$
The left boundary index is 1 and the right boundary index is 1.
Median index $\lfloor 1+1 \rfloor = 1$, the median number is $-5259$.
The target number $-5259$ is equal to the median number $-5259$, the search ends.
The target number $-5259$ is located at index $\boxed{1}$.

Figure D4: Example of TAIL-CoT-*styled*, improving human readability on top of core modules

# E    DATA PROPORTION STUDY

For each task, we divided the data into three length ranges (S, M, and L), and synthesized 100,000 training samples for each range. In studying length generalization, we trained solely on the S-range data without including any M- or L-range samples (*i.e.*, the data proportion is `<1:0:0>`), and then evaluated on all three ranges to assess the ability to generalize to longer sequences. In this section, for tasks that haven't reached saturation, we progressively incorporate longer sequences into the S-range training data and investigate the data proportion that achieves saturation performance with the minimal number of training tokens. Specifically, we keep the total number of training samples fixed, while varying the proportions of the three length ranges as `<1:0:0>`, `<8:1:1>`, `<7:2:1>`, `<5:3:2>`, and `<4:3:3>`.

As shown in Figure 6, we found that using TAIL-CoT, performance saturation can be quickly achieved by simply *adding a small amount of long data* to a large amount of short data. Guided by this observation, TAIL-CoT can leverage imbalanced sequence length proportions to reduce training costs. We call this **length generalization activation**.

# F    TASK LENGTH RANGE

| Algorithm | Task Name | Small (S) | Medium (M) | Long (L) |
|---|---|---|---|---|
| Simulation | Large Number Addition | $[10, 30]$ | $[31, 40]$ | $[41, 50]$ |
| | Large Number Division | $[2, 5]$ | $[6, 10]$ | $[11, 20]$ |
| | Bubble Sort | $[2, 4]$ | $[5, 6]$ | $[7, 8]$ |
| | Word Flip | $[10, 20]$ | $[21, 50]$ | $[51, 100]$ |
| | Compare Numbers | $[5, 10]$ | $[11, 20]$ | $[21, 50]$ |
| Recursion | Any Substring | $[3, 5]$ | $[6, 9]$ | $[10, 14]$ |
| | Derangement | $[3, 30]$ | $[31, 60]$ | $[61, 100]$ |
| Iteration | Population Growth | $[1, 10]$ | $[11, 25]$ | $[26, 50]$ |
| Greedy | Dijkstra | $[3, 5]$ | $[6, 10]$ | $[11, 20]$ |
| | Multi-Machine Schedule | $[5, 10]$ | $[11, 20]$ | $[21, 50]$ |
| Enumeration | Count Letters | $[2, 6]$ | $[7, 10]$ | $[11, 20]$ |
| | Diophantine Equation | $[10, 50]$ | $[51, 100]$ | $[101, 200]$ |
| | Prime Search | $[5, 100]$ | $[101, 200]$ | $[201, 300]$ |
| DP | 0-1 Knapsack | $[2, 3]$ | $[4, 5]$ | $[6, 8]$ |
| | LCS | $[2, 6]$ | $[7, 9]$ | $[10, 12]$ |
| | Levenshtein Distance | $[2, 4]$ | $[5, 7]$ | $[8, 9]$ |
| Divide & Conquer | Binary Search | $[5, 20]$ | $[21, 40]$ | $[41, 70]$ |
| Backtracking | Palindrome Combination | $[2, 4]$ | $[5, 6]$ | $[7, 8]$ |

Table F1: The setting of length ranges across all tasks. See Table C1 for the definitions of *length*.

# G    BASELINE DATA CONSTRUCTION

This section describes the data construction methods for the baseline methods (*i.e.*, Index Hint and Reversed Format). The experimental task was Large Number Addition (Simulation algorithm), and unlike the experiments in previous works, in this paper we contain a random number of decimals. We followed data construction methods accordingly based on the principles of these baselines.

## G.1    INDEX HINT

The Index Hint(Zhou et al., 2023; 2024) method refers to adding a hint of indexes to the corresponding numeric or logical bits of two operands for positioning in arithmetic or parity operations. This method has been extensively proven to be effective in both tasks. To compare the performance of

Index Hint and TAIL, we refer to the method (Zhou et al., 2024) that displays the indexes to locate as follows:

<p align="center">3a6b1c+5a7b6c=9a3b7c</p>

However, the above approach is for the case where two operands have the same number of digits without decimals, so we make following improvements for non-fixed-length cases with decimals:

<p align="center">1(-c)2(-b)3(-a).4(a)5(b)+6(-b)7(-a).8(a)9(b)
=1(-c)9(-b)1(-a).3(a)4(b)</p>

## G.2 REVERSED FORMAT

Reversed Format (Zhou et al., 2024; Lee et al., 2023; Shen et al., 2023; Zhou et al., 2023; Martínez et al., 2023; McLeish et al., 2024) refers to reversing each of the two operands in arithmetic operations such as addition. The rationale for this method is that the addition is usually performed from the first digit, *i.e.*, from right to left. However, the order of next token prediction (NTP) in large language models (LLMs) is from left to right, which leads to overly complex search paths during model learning and affects the length generalization performance. This method is also a widely proven effective length generalization facilitation method, constructed as follows:

<p align="center">(Origin) 123.45+67.89=191.34
(Reversed Format) 54.321+98.76=43.191</p>

## H DETAILS ABOUT MEMORY FETCHER

This section presents the details of the Memory Fetcher in TAIL. As shown in Figure 3, the Memory Fetcher is designed to ***decouple*** long-range attention from the reasoning action. It first retrieves all relevant operands from the long sequence to the end, and then performs more precise reasoning through local attention.

We visualize this local attention and compare it with the traditional CoT. As shown in Figure H1, attention across layers tends to focus on the end of the sequence. With the introduction of the Memory Fetcher, operands are captured more accurately. In contrast, traditional CoT must simultaneously attend to reasoning actions (via local attention) and the retrieval of distant operands (via long-range attention), which results in a significant sparsification of long-range attention.

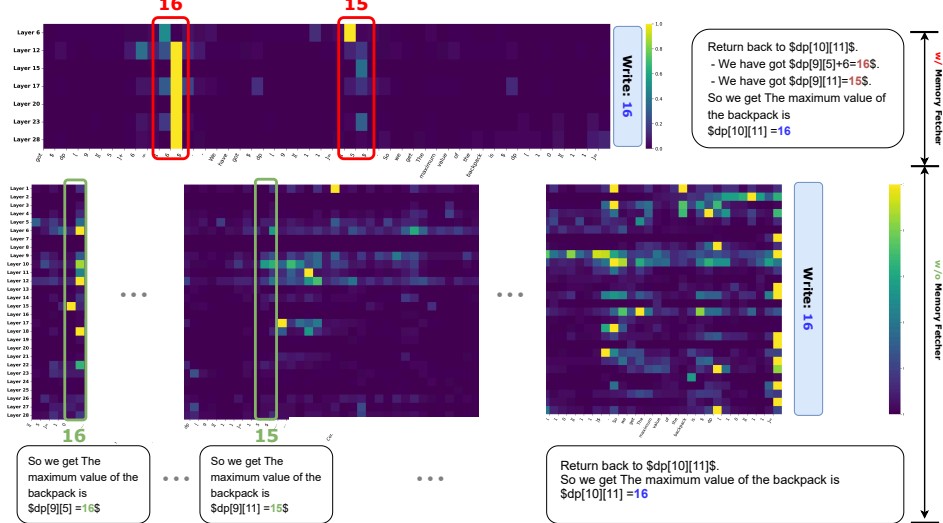

Figure H1: Ablation study on attention visualization of Memory Fetcher.

# I    COMBINATORIAL GENERALIZATION RESULTS

For tasks that belong to the same algorithmic idea, we test whether they have combinatorial generalization (*i.e.*, generalization between tasks). As shown in Figure I1, the combinatorial generalization property is not significant, which is the target of our future works.

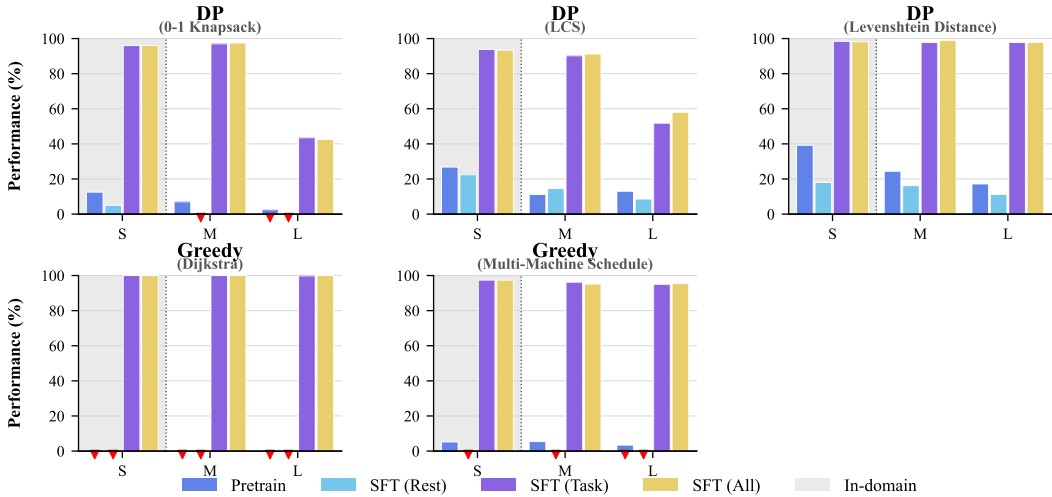

Figure I1: Generalization performance between tasks within a single algorithm (*e.g.*, DP and Greedy). **Pretrain** represents Qwen 2.5-7B as the basis for subsequent SFT. For each task, **SFT(Rest)** indicates training using data from other tasks within the algorithm, **SFT(Task)** indicates training using data from this task, and **SFT(All)** indicates training using data from all tasks within this algorithm. ▼ indicates that this piece of data has a label accuracy of less than 5%.

# J    CoT LENGTH COMPARISON

Since TAIL simulates the implementation of a Turing machine, where all algorithm steps are expanded and explicit recall of operands is added, it may lead to a significant increase in CoT. Therefore, taking *Compare Number* task of *Simulation* algorithm as an example, we compared the average lengths of TAIL-CoT and DeepSeek-R1 CoT.

As shown in Table J1, despite the increased length, the average number of tokens of TAIL remains on par with that of DeepSeek-R1. With such acceptable length, it significantly outperforms the 671B reasoning model with only 7B parameters, which demonstrates the efficiency of TAIL.

| Metric | TAIL-CoT (7B, Finetuned Qwen2.5 7B) | DeepSeek-R1 (671B) |
|---|---|---|
| avg. Tokens | 1455 | 1461 |
| Label Accuracy | 90.0 | 51.2 |

Table J1: Comparison between TAIL-CoT (7B, Finetuned Qwen2.5 7B) and DeepSeek-R1 (671B) on average token usage and label accuracy. The average number of tokens retains the integer part.

# K    DEEPSEEK-R1-DISTILL TRAINING RESULTS

For a fairer comparison, we fine-tuned Qwen2.5-7B using the same amount of DeepSeek-R1-Distill data and TAIL-CoT. As shown in Table K1, although DeepSeek-R1's CoT can significantly improve in-domain performance, the improvement is not as significant as TAIL-CoT, and the performance shows a sharp decline in longer sequences. Since DeepSeek-R1 has a low accuracy rate on some tasks, distilling the same amount of training data requires a large number of tokens, so this experiment was not conducted on a large scale.

| Setting | TAIL (7B) | R1-Distill-Qwen2.5-7B |
|---|---|---|
| S (In Domain) | 98.0 | 72.2 |
| M | 94.2 | 67.2 |
| L | 90.0 | 61.8 |

Table K1: Performance comparison between TAIL (7B) and R1-Distill-Qwen2.5-7B across different sequence lengths. (**S** = Short sequence data, **M** = Medium sequence data, **L** = Long sequence data)

## L    COT STRUTURE COMPARISON OF DEEPSEEK-R1 AND TAIL

Take a *Word Flip* task of *Simulation* algorithm as an example.

**Zero-shot Query**: Provide the flipped string based on "As he measured, he said, Every Ollivander wand has a core of a powerful magical substance". Reverse the entire string character by character (including spaces).

### L.1    RESPONSE OF DEEPSEEK-R1

```
<think>
We are to flip the string based on the given sentence: "As he measured,
    he said, Every Ollivander wand has a core of a powerful magical
    substance"
The instruction is to "flip" the string. Let me reverse the entire string
    :
Original: "As he measured, he said, Every Ollivander wand has a core of a
     powerful magical substance" We can reverse it character by character
    .
Steps:
1. Take the string.
2. Reverse the order of the characters.
Example: "hello" becomes "olleh"
Let's do it:
Original: 'A','s',' ','h','e',' ','m','e','a','s','u','r','e','d',',',' '
    ',','h','e',' ','s','a','i','d',',',',',' ','E','v','e','r','y',' ','O',' l
    ','l','i','v','a','n','d','e','r',' ','w','a','n','d',' ','h','a','s
    ',' ',' ','a',' ',' ','c','o','r','e',' ',' ','o','f',' ',' ','a',' ',' ','p','o','w','e
    ',' ','r','f','u','l',' ',' ','m','a','g','i','c','a','l',' ',' ','s','u','b','s
    ',' ','t','a','n','c','e'
Reverse:
Start from the last character and go to the first.
But note: we can use a simple method in programming to reverse the string
    .
However, since we are to do it manually, I'll write the reversed string.
Let me break the original string into parts and then reverse the whole:
Original: "As he measured, he said, Every Ollivander wand has a core of a
     powerful magical substance"
We can write the reversed string as:
Last character: 'e' Then: 'c', 'n', 'a', 't', 's', 'b', 'u', 's', ' ', 'l
    ', 'a', 'c', 'i', 'g', 'a', 'm', ' ', 'l', 'u', 'f', 'r', 'e', 'w', '
    o', 'p', ' ', 'a', ' ', 'f', 'o', ' ', 'e', 'r', 'o', 'c', ' ', 'a',
    ' ', 's', 'a', 'h', ' ', 'd', 'n', 'a', 'w', ' ', 'r', 'e', 'd', 'n',
     'a', 'v', 'i', 'l', 'l', 'O', ' ', 'y', 'r', 'e', 'v', 'E', ' ',
    ',', 'd', 'i', 'a', 's', ' ', 'e', 'h', ' ', ',', 'd', 'e', 'r', 'u',
     'a', 's', 'e', 'm', ' ', 'e', 'h', ' ', 's', 'A'
But wait, let's do it step by step:

...... (A lot of splitting to achieve the number of digits omitted.)

Wait, I think I made a mistake.
Let me write the entire string without spaces and then add the spaces and
     punctuation:
Actually, it's easier to just reverse the entire string as a whole.
```

```
We can write the string as: s = "As he measured, he said, Every
    Ollivander wand has a core of a powerful magical substance"
Reversed string = s[::-1] in Python.
But manually:
Start from the end: 'e' (from substance) 'c' 'n' 'a' 't' 's' 'b' 'u' 's'
    ' ' (space before substance?
but actually the last word is "substance", and before that is "magical",
    so there's a space between them)

...... (A lot of repeated verification attempts omitted.)

Actually, the string is: ... a powerful magical substance So the order in
    the original is: ... 'a', space, 'p','o','w','e','r','f','u','l',
    space, 'm','a','g','i','c','a','l', space, 's','u','b','s','t','a','n
    ','c','e' Therefore, when reversed, we start from the last character
    of the last word and go backwards: Start: 'e' Then: 'c','n','a','t','
    s','b','u','s' -> then space -> then the word "magical" reversed: 'l
    ','a','c','i','g','a','m' -> then space -> then the word "powerful"
    reversed: 'l','u','f','r','e','w','o','p' -> then space -> then 'a'
    -> then space -> then the word "of" becomes 'f','o'?

...... (A lot of attempts at different reversal methods omitted.)

Actually, the comma we already handled? Wait, let me reconstruct: The
    original string: "As he measured, he said, Every Ollivander wand has
    a core of a powerful magical substance" Breaking down: "As" -> ['A','
    s'] space -> [' '] "he" -> ['h','e'] space -> [' '] ...

We can reverse the entire list. The last character is 'e' (from substance
    ), then the one before is 'c', then 'n', then 'a', then 't', then ...

Therefore, the flipped string is: "ecnatsbus lacigam lufrewop a fo eroc a
    sah dnaw rednavillO yrevE ,dias eh ,derusaem e h sA" But note: the
    original had a space at the end? No, the last word is "substance"
    without any trailing space. Similarly, the first word "As" has no
    leading space. So the reversed string should not have a leading space
    ? It ends with "sA", which is correct. Final answer: "ecnatsbus
    lacigam lufrewop a fo eroc a sah dnaw rednavillO yrevE ,dias eh ,
    derusaem e h sA"
</think>

The original string is:
"As he measured, he said, Every Ollivander wand has a core of a powerful
    magical substance"

Reversing the entire string character by character (including spaces,
    punctuation, and maintaining case) results in:
**"ecnatsbus lacigam lufrewop a fo eroc a sah dnaw rednavillO yrevE ,dias
    eh ,derusaem e h sA"**

### Explanation:
- The reversal is done on a per-character basis, so the last character of
    the original string ("e" from "substance") becomes the first
    character of the flipped string, and so on.
- Spaces, commas, and capitalization are preserved as they appear in the
    original.
- This is not a word reversal (where words are reversed but characters
    within words stay in order); it is a full character-level reversal.
```

The final output is "ecnatsbus lacigam lufrewop a fo eroc a sah dnaw rednavillO yrevE ,dias eh ,derusaem e h sA". The model tried many methods, but did not actually perform the flipping operation step by step, and directly output an incorrect result at the end of the inference.

## L.2 TAIL-CoT-*styled*

```
<think>
Decompose the string bit by bit(starting from $1$):
 - Index $1$: 'A'
 - Index $2$: 's'
 - Index $3$: ' '
 - Index $4$: 'h'
 - Index $5$: 'e'
 - Index $6$: ' '
 - Index $7$: 'm'
 - Index $8$: 'e'
 - Index $9$: 'a'
 - Index $10$: 's'

 .... (Omitted and will be expanded in actual reasoning.)

 - Index $80$: ' '
 - Index $81$: 's'
 - Index $82$: 'u'
 - Index $83$: 'b'
 - Index $84$: 's'
 - Index $85$: 't'
 - Index $86$: 'a'
 - Index $87$: 'n'
 - Index $88$: 'c'
 - Index $89$: 'e'

We find that 'As he measured, he said, Every Ollivander wand has a core
    of a powerful magical substance' has $89$ words, indexed from $1$ to
    $89$.

The original string is the concatenation of the letters with indices from
     $1$ to $89$. If you reverse it, the result is the reverse
     concatenation of the indices, that is, from $89$ to $1$.

Let's perform the reverse stitching step by step:
 - Index $89$: 'e', intermediate result is 'e'
 - Index $88$: 'c', intermediate result is 'ec'
 - Index $87$: 'n', intermediate result is 'ecn'
 - Index $86$: 'a', intermediate result is 'ecna'
 - Index $85$: 't', intermediate result is 'ecnat'
 - Index $84$: 's', intermediate result is 'ecnats'
 - Index $83$: 'b', intermediate result is 'ecnatsb'
 - Index $82$: 'u', intermediate result is 'ecnatsbu'
 - Index $81$: 's', intermediate result is 'ecnatsbus'
 - Index $80$: ' ', intermediate result is 'ecnatsbus '

 .... (Omitted and will be expanded in actual reasoning.)

 - Index $10$: 's', intermediate result is 'ecnatsbus lacigam lufrewop a
     fo eroc a sah dnaw rednavillO yrevE ,dias eh ,derus'
 - Index $9$: 'a', intermediate result is 'ecnatsbus lacigam lufrewop a
     fo eroc a sah dnaw rednavillO yrevE ,dias eh ,derusa'
 - Index $8$: 'e', intermediate result is 'ecnatsbus lacigam lufrewop a
     fo eroc a sah dnaw rednavillO yrevE ,dias eh ,derusae'
 - Index $7$: 'm', intermediate result is 'ecnatsbus lacigam lufrewop a
     fo eroc a sah dnaw rednavillO yrevE ,dias eh ,derusaem'
 - Index $6$: ' ', intermediate result is 'ecnatsbus lacigam lufrewop a
     fo eroc a sah dnaw rednavillO yrevE ,dias eh ,derusaem '
 - Index $5$: 'e', intermediate result is 'ecnatsbus lacigam lufrewop a
     fo eroc a sah dnaw rednavillO yrevE ,dias eh ,derusaem e'
 - Index $4$: 'h', intermediate result is 'ecnatsbus lacigam lufrewop a
     fo eroc a sah dnaw rednavillO yrevE ,dias eh ,derusaem eh'
```

```
 - Index $3$: ' ', intermediate result is 'ecnatsbus lacigam lufrewop a
     fo eroc a sah dnaw rednavillO yrevE ,dias eh ,derusaem eh '
 - Index $2$: 's', intermediate result is 'ecnatsbus lacigam lufrewop a
     fo eroc a sah dnaw rednavillO yrevE ,dias eh ,derusaem eh s'
 - Index $1$: 'A', intermediate result is 'ecnatsbus lacigam lufrewop a
     fo eroc a sah dnaw rednavillO yrevE ,dias eh ,derusaem eh sA'

</think>

The final result is 'ecnatsbus lacigam lufrewop a fo eroc a sah dnaw
    rednavillO yrevE ,dias eh ,derusaem eh sA'.
```

