# OpenReview forum: "The Imitation Game: Turing Machine Imitator is Length Generalizable Reasoner"
_ICLR.cc/2026/Conference — ICLR 2026 Poster_

### Official Review · Reviewer_txZ2 · 2025-10-25

**Soundness:** 4
**Presentation:** 3
**Contribution:** 3
**Rating:** 6
**Confidence:** 5

**Summary:**

This paper presents TAIL, which is a specific chain-of-thought format inspired by Turing machines that facilitates length generalization. It synthesizes large datasets for a set of traditional algorithms and then finetunes Qwen models on that data and illustrates length generalization exceeding standard reasoning models (R1).

**Strengths:**

1. Compared to prior work on length generalization it considers a more varied set of tasks and shows strong results.
2. Most prior work on length generalization focuses on small models trained from scratch on simple tasks, this instead finetunes large models in a way that is perhaps more interesting or has more practical implications.
3. The specific method for generating varied synthetic data in the Turing machine style is clever.
4. The ablations of the chain-of-thought format are rigorous and useful

**Weaknesses:**

1. The paper over-claims the novelty of the idea of using a Turing-machine formulated CoT for length generalization. This is the central point of Hou et al. which is barely cited in the related work section. This paper does do much larger scale experiments and does study some intricacies of how to exactly define the CoT when finetuning larger models, but the core idea is exactly the same as Hou et al. and this should be made more clear in the paper.
2. Some of the claims of universality should also be more careful. While it is true that hypothetically any algorithmic task can be formulated as a TAIL-style cot, this still requires someone to hand-design the linearization, state, and memory fetcher components of the CoT used for the SFT data.

**Questions:**

none

---

> ### Author Response · Authors · 2025-11-27
> **Appreciation for Your Review and Responses to Raised Concerns**
>
> Thank you very much for your positive feedback and for the valuable comments. Below, we address the concerns you raised, and we hope our clarifications will help resolve any remaining doubts.
>
> ***
>
> ## **Response to Weakness 1 about Novelty**
>
> Thank you for raising this point. In the official comment `Summary Response on the Motivation and Contributions` of TAIL (at the top of the page), we provide a detailed explanation of the unique motivation and contributions of this work.
>
> We acknowledge that Hou et al. share similarities with the core idea of our work. However, as discussed in Lines 447–449 of the manuscript, TAIL differs from Hou et al. in that it is not restricted to Hard-ALiBi positional encodings and is evaluated across a much broader set of tasks. More importantly, the contribution of TAIL lies in identifying the two fundamental causes of length generalization failure: (1) **shortcut CoT patterns** (addressed by *Atomic State* and *Linear Transition*) and (2) **sparse long-distance attention** (addressed by *Memory Fetcher*). This allows us to pinpoint the key CoT modules essential for length generalization, supported by extensive ablation studies.
>
> Thank you again for pointing this out! We will add this discussion in the revised manuscript.
>
> ***
>
> ## **Response to Weakness 2 about Universality**
>
> We acknowledge that parts of our writing may appear overclaimed, and we will carefully refine the wording in the revised manuscript. However, we would like to clarify that the notion of **universality** in our paper refers to the ability of our method to apply to **any computable problem**, whereas prior approaches such as index hints or reversed formats cannot be used to systematically construct data for a wider range of tasks. To validate the underlying programmatic idea, we conducted experiments on **18 tasks spanning 8 major algorithmic classes**, demonstrating strong generalization across all of them.
>
> ***
>
> Thank you again for your valuable comments, and we look forward to your response.

---

### Official Review · Reviewer_8yiH · 2025-10-29

**Soundness:** 3
**Presentation:** 2
**Contribution:** 1
**Rating:** 2
**Confidence:** 3

**Summary:**

This paper proposes TAIL, a data-driven framework for improving length generalization in LLMs. By synthesizing cot data that mimics the execution of a Turing machine, TAIL improves the performance of LLM's ability of length generalization. In general, TAIL enforces three structural properties, linear transition, atomic state and memory fetcher.

The method is evaluated on a challenging synthetic dataset spanning 8 algorithmic classes and 18 tasks. Results show that TAIL significantly enhances length generalization in Qwen2.5-7B, outperforming prior task-specific methods and even larger models like DeepSeek-R1.

**Strengths:**

1. The Turing machine alignment provides a principled approach to structured reasoning, systematically addressing length generalization through linearized execution and explicit memory management.
2. Evaluation across 8 algorithm classes and 18 tasks demonstrates impressive universality beyond simple tasks commonly studied.

**Weaknesses:**

1. Incomplete baselines. The paper doesn't compare against standard CoT fine-tuning, making it unclear whether gains come from Turing imitation or simply SFT. Comparisons are limited to un-trained base models.

2. The approach requires manually writing Python code for each task to generate TAIL-formatted CoT data, creating significant engineering overhead. This limits real-world adoption where users need automated solutions for new tasks.

3. The paper overlooks important connections to relevant literature. Two papers "Show Your Work: Scratchpads for Intermediate Computation with Language Models", "Beyond Single-Task: Robust Multi-Task Length Generalization for LLMs" deserves explicit discussion. The latter explores similar concepts of program-based CoT generation for length generalization but extends to multi-task transfer learning.

**Questions:**

1. Can the authors provide more baselines such as SFT with simple CoT or SFT without CoT?
2. It seems that simulating Turing execution with atomic states and memory fetching substantially increases reasoning length. Is it true? If so, will it be limited by the finite context, causing TAIL to actually fail in solving very long problems?
3. What are the additional contributions and innovations of TAIL compared to the two works mentioned above?

---

> ### Author Response · Authors · 2025-11-27
> **Appreciation for Your Review and Responses to Raised Concerns (Response-1)**
>
> Thank you very much for your thoughtful and valuable comments; we truly appreciate them. Below, we provide our responses to your concerns, and we hope these clarifications will be helpful.
>
> ***
>
> ## **Response to Weakness 1 and Question 1 about Baselines**
>
> Thank you for raising this question. Appendix J includes results after standard CoT SFT, where **both label accuracy and length generalization are significantly worse than TAIL**. Specifically, we use DeepSeek-R1–distilled correct responses as training data and perform SFT on Qwen2.5-7B under identical conditions (same number of samples and hyperparameters). The results are shown in Table J1 (copied below).
>
> | Setting           | TAIL (7B) | R1-Distill-Qwen2.5-7B |
> |-------------------|-----------|-------------------------|
> | S (In Domain)     | 98.0      | 72.2                    |
> | M                 | 94.2      | 67.2                    |
> | L                 | 90.0      | 61.8                    |
>
> > **Table J1:** Performance comparison between TAIL (7B) and R1-Distill-Qwen2.5-7B across different sequence lengths. (S = Short sequence data, M = Medium sequence data, L = Long sequence data)
>
> Because DS-R1 has low accuracy on some tasks, collecting an equal number of correct responses is costly and time-consuming. So this experiment is conducted on a limited subset. We will highlight this in the revised manuscript and add an explicit pointer to the appendix.
>
> ***
>
> ## **Response to Weakness 2 about Engineering Overhead**
>
> Your understanding is correct that TAIL requires manually writing new code for each task. However, this paper focuses on challenging computable tasks, and writing a program is currently a **relatively low-cost way** to obtain CoT data:
> - Manual annotation is expensive
> - Distilling advanced LLMs has low accuracy, requiring several times more API calls and time to collect the same amount of correct responses for training
> - TAIL requires writing only a single program, which can then generate hundreds of thousands of training examples efficiently through program execution.
>
> We acknowledge that real world adoption is currently limited (as noted in Appendix L). Nevertheless, we demonstrate unprecedented performance on individual tasks, and future work will extend TAIL to more practical scenarios through compositional generalization (as you noted, Meta-RFFT provides valuable inspiration for our future research on compositional generalization).
>
> ***
>
> ## **Response to Question 2 about reasoning length**
>
> Thank you for pointing this out. We have considered this concern, and our experiments show that although TAIL increases the CoT length, it remains significantly shorter than that of reasoning models such as DeepSeek-R1.
>
> As shown in Appendix I (table copied below), TAIL uses a **slightly shorter average reasoning token count** than DeepSeek-R1, yet achieves approximately a **40% performance improvement**.
>
> | Metric          | TAIL-CoT (7B, Finetuned Qwen2.5 7B) | DeepSeek-R1 (671B) |
> |-----------------|--------------------------------------|----------------------|
> | avg. Tokens     | 1455                                 | 1461                 |
> | Label Accuracy  | 90.0                                 | 51.2                 |
>
> > **Table I1:** Comparison between TAIL-CoT (7B, Finetuned Qwen2.5 7B) and DeepSeek-R1 (671B) on average token usage and label accuracy. The average number of tokens retains the integer part.
>
> Although TAIL expands the reasoning process and introduces multiple modules, we remove many human-like thinking styles and **retain only the core symbolic computation content**, which helps keep the CoT length controlled. The ablation results further show that removing these styles does not harm performance (Figure 4). This supports our claim that the essence of length generalization lies in TAIL’s core modules rather than in human-like styles.

---

> > ### Author Response · Authors · 2025-11-27
> > **Appreciation for Your Review and Responses to Raised Concerns (Response-2)**
> >
> > ### **Response to Weakness 3 and Question 3 about Relevant Literature**
> >
> > We apologize for overlooking these important papers, and we appreciate you pointing them out!
> >
> > In the official comment `Summary Response on the Motivation and Contributions of TAIL` (at the top of the page), we provide a detailed explanation of the motivation and contributions of our manuscript. Below, we outline the key innovations of TAIL relative to these two papers:
> >
> > 1. Nye et al., 2021 introduces “scratchpad”, enabling Transformers to perform complex multi-step algorithms and improving reasoning over longer sequences. However, TAIL can be viewed as an upgraded form of scratchpad, it first **identifies the underlying causes of length generalization failure** (shortcut reasoning and sparse long-range attention), and then **introduces key CoT modules** specifically designed to address these issues. Moreover, TAIL is validated on a much broader range of problems, covering all major algorithm families.
> > 2. Hu et al., 2025 proposes Meta-RFFT, a method focused on multi-task mixed length generalization. It introduces a robustness-enhancing training strategy that enables models to achieve stable and cross-task length generalization, with impressively strong performance across a wide range of tasks. However, TAIL conducts a detailed analysis of **why** length generalization fails and introduces core CoT modules that address these root causes, achieving strong length generalization on **individual high-difficulty algorithmic tasks**.
> >
> > Thank you again for pointing this out. We will **make sure** to add these references in the revised version.

---

### Official Review · Reviewer_RUTK · 2025-10-30

**Soundness:** 3
**Presentation:** 3
**Contribution:** 3
**Rating:** 4
**Confidence:** 4

**Summary:**

This paper focuses on a key limitation of current large language models: length generalization, meaning their ability to maintain correct reasoning on inputs far longer than those encountered during training. To address this, the authors propose TAIL (Turing mAchine Imitation Learning), a training approach let the mode mimic the step-by-step execution of a Turing machine. Experiments show that using only this synthetic data, TAIL can improve model’s length generalization ability and task performance.

**Strengths:**

1. This paper is well-motivated, studying  length generalization – an important problem of LLM
2. The proposed method only require synthetic data, which makes it easier to extend to more tasks.
3. The paper performed comprehensive experiments and achieved improvements across diverse benchmarks, demonstrating strong generality and robustness.

**Weaknesses:**

1. Limited practical significance: The benchmarks are largely rely on deterministic symbolic computation. The so-called Chain-of-Thought effectively becomes a **Chain of Computations**, simply executing predefined procedural steps rather than engaging in genuine high-level reasoning. As a result, the improvements may reflect better simulation of algorithmic traces, rather than any substantive enhancement in “reasoning ability.”
2. Limited Transferability to real-world reasoning scenarios. The work does not evaluate performance on realistic tasks that require conceptual decision-making or deeper inference, such as multi-hop mathematical reasoning or logic problems. So it is unclear how TAIL can contribute to practical reasoning challenges.

**Questions:**

1. How does the proposed method extend to more realistic reasoning domains, such as GSM8K, MATH, or BIG-bench reasoning tasks? It would be crucial for verifying the real effectiveness of proposed method.
2. The work argues that TAIL improves computational reasoning, but many of the evaluated tasks could be directly solved LLM's tool use. Why just let LLM to write the code and execute the program for these type of tasks with deep computation depths? What advantages does TAIL provide over the agentic method?

---

> ### Author Response · Authors · 2025-11-27
> **Appreciation for Your Review and Responses to Raised Concerns**
>
> Thank you for the valuable feedback; it will greatly help us improve the manuscript. Below, we address your concerns and hope these clarifications resolve them.
>
> ***
>
> ## **Response to Weakness 1 about Chain of Computations**
>
> Your understanding is correct, TAIL is indeed essentially a “chain of computations” composed of deterministic symbolic operations. However, the scope of this paper is limited to **computable problems under deterministic algorithms** (as noted in Line 75 of Intro). Our goal is not to demonstrate human-like high-level reasoning, but to systematically study the length generalization behavior of LLMs on this class of problems.
>
> In addition, the “reasoning ability” discussed in this paper refers to **reliably executing algorithms on computable tasks and extrapolating stably to longer lengths**, rather than general human-like intuitive high-level reasoning. Specifically, human-like shortcut tendencies can cause the model to skip steps or repeatedly try multiple approaches without step-by-step deep reasoning (a pattern clearly observed in traditional CoT; see an example of DS-R1 response in `Appendix K.1`). These behaviors prevent the model from handling longer sequences [1–2]. We have tried performing SFT using distilled CoTs that exhibit “human-like thinking”, but resulted in poor length generalization (see `Appendix J`). Moreover, removing the human-like thinking style in the ablation study (see `Figure 4`) does not degrade performance. These results suggest that focusing solely on human reasoning patterns may not be the key to improving length generalization.
>
> Therefore, enabling the model to simulate algorithmic execution and achieve length generalization goes beyond surface-level pattern matching. It leads to **substantive changes in the internal attention distribution** within Transformer layers (see attention maps in `Appendix G`). For more details on the motivation and analysis behind TAIL’s length generalization, please refer to the official comment `Summary Response on the Motivation and Contributions of TAIL` at the top of the page.
>
> ***
>
> ## **Response to Weakness 2 and Question 1 about Transferability to Real-world Scenarios**
>
> Your understanding is correct. Generalizing to real-world tasks is indeed a current limitation (as noted in `Appendix L`). We have also evaluated TAIL on MATH, and it does not yield improvements at this stage.
>
> Realistic tasks typically involve compositions of multiple reasoning subtasks. In this paper, we focus on providing a universal CoT framework that substantially improves the reasoning performance of **individual computable tasks**, as measured through length generalization. Extending TAIL to handle **compositional generalization across multiple task** is an important direction for future work.
>
> ***
>
> ## **Response to Question 2 about Comparison to LLM's Tool Use**
>
> The goal of TAIL is not to compete with tool-based approaches, but to address the question of whether an LLM’s **intrinsic computational generalization ability** can be systematically improved through structured CoT.
>
> Advantages of TAIL over tool-based approaches:
> - No need for formalizing problems into code, TAIL can be applied directly via SFT on existing pretrained LLMs.
> - No reliance on external systems (e.g., sandboxes), resulting in minimal environmental dependencies.
> - Ability to internalize multiple tool-like capabilities (e.g., code interpreter, calculator) within a single semantic space, with the goal of improving the model’s intrinsic performance.
>
> ***
>
> ### **Reference**
>
> [1] Anil C, Wu Y, Andreassen A, et al. Exploring length generalization in large language models[J]. Advances in Neural Information Processing Systems, 2022, 35: 38546-38556.
>
> [2] Saparov A, Pawar S, Pimpalgaonkar S, et al. Transformers struggle to learn to search[J]. arXiv preprint arXiv:2412.04703, 2024.

---

### Official Review · Reviewer_P9RF · 2025-10-31

**Soundness:** 2
**Presentation:** 2
**Contribution:** 2
**Rating:** 2
**Confidence:** 3

**Summary:**

This paper proposes Turing Machine Imitation Learning (TAIL), which synthesizes CoT data that imitates the execution process of a Turing Machine to enhance the length generalization of Transformer-based LLMs. The authors claim that fine-tuning a model on synthetic data generated via TAIL improves its length generalization capability. They validate this claim on Qwen2.5-7B across various tasks and show that the fine-tuned model consistently outperforms the original one.

**Strengths:**

1. The idea of imitating Turing Machine execution is interesting and conceptually appealing.

2. TAIL is orthogonal to other approaches such as Index Hint, making it easily combinable with them.

**Weaknesses:**

1. The paper lacks sufficient elaboration and analysis on why TAIL improves length generalization.

2. The experimental design does not convincingly demonstrate TAIL’s effectiveness. Specifically, the comparison between the original Qwen2.5-7B and the model fine-tuned on task-specific data is not fair. Moreover, the paper does not include comparisons with other prompt-engineering methods, such as Program-of-Thought, to justify the claimed advantages of TAIL.

**Questions:**

See Weaknesses.

---

> ### Author Response · Authors · 2025-11-27
> **Appreciation for Your Review and Responses to Raised Concerns**
>
> Thank you for the valuable feedback which is highly helpful for improving our manuscript. Below we provide responses to the raised concerns, and we hope these clarifications address them.
>
> ***
>
> ## **Response to Weakness 1 about Effectiveness Analysis**
>
> We apologize that, due to space limitations, our explanation in the manuscript was not sufficiently clear.
>
> Therefore, in the official comment `Summary Response Regarding the Contributions of TAIL` (on the top of the page), we provide a clearer analysis. In short, we first identify the root causes of poor length generalizaiton performance: **shortcut reasoning pattern** [1–2] and **sparse long-distance attention** in reasoning actions [3]. We then present targeted solutions: Atomic State and Linear Transition address shortcut behavior at the micro and macro levels, respectively; Memory Fetcher resolves the sparse-attention issue. The attention maps further illustrate that the performance gains stem from a change in the underlying attention pattern.
>
> We ensure that this analysis will be explained more clearly in the next version of the manuscript.
>
> ***
>
> ## **Response to Weakness 2 about Insufficient Comparison**
>
> In the main experiments of the manuscript, we indeed only compared against the original Qwen2.5-7B. Below, we provide additional results for a more fair and comprehensive comparison:
>
> ### **1. Comparison with Standard CoT SFT**
>
> Appendix J of the manuscript already includes results after standard CoT SFT. Both the label accuracy and the length generalization performance are significantly worse than those of TAIL.
>
> Specifically, we use the DeepSeek-R1–distilled responses filtered for correctness as the training data. Under identical experimental conditions (same number of training samples and identical training hyperparameters), we perform SFT on Qwen2.5-7B. The results are shown in Table J1 (copied below).
>
> | Setting           | TAIL (7B) | R1-Distill-Qwen2.5-7B |
> |-------------------|-----------|-------------------------|
> | S (In Domain)     | 98.0      | 72.2                    |
> | M                 | 94.2      | 67.2                    |
> | L                 | 90.0      | 61.8                    |
>
> > **Table J1:** Performance comparison between TAIL (7B) and R1-Distill-Qwen2.5-7B across different sequence lengths. (S = Short sequence data, M = Medium sequence data, L = Long sequence data)
>
> Because DS-R1 has relatively low accuracy on several tasks, collecting the same number of correct responses requires substantial cost and time. So this experiment is conducted only on a limited subset. We will emphasize this point in the revised manuscript and add a clear pointer to appendix J.
>
> ### **2. Comparison with Prompt-engineering**
>
> Program-of-Thought [4] is essentially a form of external tool usage (a code interpreter). It requires sandbox construction and verification of code correctness. In contrast, TAIL aims to improve the model’s intrinsic high-level reasoning ability purely through SFT.
>
> We also provide prompt-engineering results that use existing TAIL-CoTs as few-shot examples (applied to the non-SFT Qwen2.5-Instruct). The results are shown in the table below.
>
> | Setting           | TAIL (7B) | Few-shot Prompt-engineering (7B) | Origin Qwen2.5-7B |
> |-------------------|-----------|-------------------------|-------------------------|
> | S (In Domain)     | 98.0      | 71.2                    | 58.4                    |
> | M                 | 94.2      | 62.2                    | 47.4                    |
> | L                 | 90.0      | 59.4                    | 38.6                    |
>
> The results indicate that prompt-engineering can substantially improve label accuracy on short sequences. However, its performance on longer sequences remains weaker than SFT, while still significantly outperforming the original model.
>
> ### **3. Summary on Comparison**
>
> SFT and prompt engineering are **two ways of applying TAIL**. Our core contribution lies in **proposing CoT modules** that promote length generalization. SFT serves as a more effective mechanism for internalizing these module-level ideas into the model’s inherent reasoning process.
>
> ***
>
> ## **Reference**
>
> [1] Anil C, Wu Y, Andreassen A, et al. Exploring length generalization in large language models[J]. Advances in Neural Information Processing Systems, 2022, 35: 38546-38556.
>
> [2] Saparov A, Pawar S, Pimpalgaonkar S, et al. Transformers struggle to learn to search[J]. arXiv preprint arXiv:2412.04703, 2024.
>
> [3] Wang R, Huang W, Song S, et al. Beyond in-distribution success: Scaling curves of cot granularity for language model generalization[J]. arXiv preprint arXiv:2502.18273, 2025.
>
> [4] Chen W, Ma X, Wang X, et al. Program of thoughts prompting: Disentangling computation from reasoning for numerical reasoning tasks[J]. arXiv preprint arXiv:2211.12588, 2022.

---

> > ### Comment · Reviewer_P9RF · 2025-11-28
> >
> > The authors have provided convincing clarifications in their rebuttal. As a result, I will adjust my score to 6.

---

### Author Response · Authors · 2025-11-27
**Summary Response on the Motivation and Contributions of TAIL**

We thank the reviewers for their time and valuable feedback. Several reviewers raised concerns about the novelty of TAIL, and noted that the paper did not sufficiently analyze why TAIL improves length generalization. Below, we provide a concise, motivation-driven summary that clarifies these points.

> `TL;DR`: To improve length generalization, TAIL resolves two fundamental limitations of traditional CoT: **shortcut reasoning** and **sparse long-distance attention**. The **Atomic State** and **Linear Transition** address shortcut behavior at the micro and macro levels respectively, while the **Memory Fetcher** mitigates the sparse-attention issue.

### **Shortcut Reasoning**

Previous works [1-2] show that when SFT'd on non-sequential CoT, LLMs prefer to take shortcuts during inference, and subsequently fail to generalize to longer sequences. From a micro perspective, shortcuts appear as **overly large single-step reasoning**, which means the model jumps to substantial sub-results instead of reasoning step by step [1-2]; From a macro perspective,  we also observe shortcut behavior as **step-skipping** and **frequent switching between reasoning strategies** (see Appendix K.1 for CoT example of DS-R1).
- **Atomic State** focuses on the micro level of each reasoning step. Guided by the RASP-L theory [4], it produces length-generalizable steps of appropriate granularity (i.e., steps without internal loops).
- **Linear Transition** focuses on the macro structure of connecting reasoning steps. It linearly unfolds the entire reasoning process step by step, and the tight logical linkage between Atomic States prevents step-skipping.

### **Sparse Long-distance Attention**
[3] shows that long-sequence reasoning leads to severe attention sparsity in later steps. As illustrated in Fig G2 (lower part), traditional CoT attends to distant past states while performing the current reasoning action, resulting in sparse long-distance attention. **Memory Fetcher** decouples long-range attention construction from action execution: (1) First build long-range attention via a simple copy operation, copying the relevant operands to the end of the sequence; (2) then performs the reasoning action.

As shown in Fig G2 (upper part), introducing Memory Fetcher significantly improves attention distribution during reasoning actions. More attention is **focused** on nearby relevant operands, improving single-step correctness, and thereby stabilizing long-sequence reasoning.

***

**Reference**

[1] Anil C, Wu Y, Andreassen A, et al. Exploring length generalization in large language models[J]. Advances in Neural Information Processing Systems, 2022, 35: 38546-38556.

[2] Saparov A, Pawar S, Pimpalgaonkar S, et al. Transformers struggle to learn to search[J]. arXiv preprint arXiv:2412.04703, 2024.

[3] Wang R, Huang W, Song S, et al. Beyond in-distribution success: Scaling curves of cot granularity for language model generalization[J]. arXiv preprint arXiv:2502.18273, 2025.

[4] Zhou H, Bradley A, Littwin E, et al. What algorithms can transformers learn? a study in length generalization[J]. arXiv preprint arXiv:2310.16028, 2023.

---

### Meta-Review · Area_Chair_4tw2 · 2026-01-10

**Summary:**

Concerns are: (i) insufficient explanation of why TAIL improves length generalization, (ii) lack of fair and complete baselines (especially standard CoT SFT and prompt-engineering methods), (iii) over-claiming of novelty and universality relative to prior work, and (iv) limited practical relevance and transferability beyond deterministic algorithmic tasks.

**Reviewer Concerns:**

Concerns addressed:
- Lack of mechanism explaining gains addressed through a detailed explanation identifying shortcut reasoning and sparse long-distance attention as root causes, and mapping them explicitly to modules. These are supported by attention analyses and ablations.
- Incomplete or unfair baselines addressed via additional comparisons with CoT SFT and few-shot baselines.
- Novelty relative to prior work addressed by clarifying differences from scratchpad-style methods and Hou et al.
- TAIL as “chain of computations” addressed by explicitly defining the scope as computable deterministic tasks and reframing “reasoning” as reliable algorithmic execution with length extrapolation.
- CoT length and context limits addressed with quantitative comparisons showing TAIL’s reasoning length is comparable to, or shorter than, DeepSeek-R1 while achieving higher accuracy.

Concerns remains:
- Limited transfer to real-world reasoning tasks
- Engineering overhead and manual program design mitigated but not eliminated.

**Reviewer Scores:**

Reviewer P9RF explicitly updated score from 2 to 6 after rebuttal.
Reviewer 8yiH: Likely increases from 2 to 3 or 4 as major baseline and literature concerns addressed.
Other reviewers are likely to remain unchanged.

---

### Decision · Program_Chairs · 2026-01-26

Accept (Poster)